# A Comparative Study of Morphology, Photosynthetic Physiology, and Proteome between Diploid and Tetraploid Watermelon (*Citrullus lanatus* L.)

**DOI:** 10.3390/bioengineering9120746

**Published:** 2022-12-01

**Authors:** Zhanyuan Feng, Zhubai Bi, Dugong Fu, Lihan Feng, Dangxuan Min, Chensong Bi, He Huang

**Affiliations:** 1Key Laboratory of Tropical Crops Germplasm Resources Genetic Improvement and Innovation of Hainan Province, Tropical Crops Genetic Resources Institute, Chinese Academy of Tropical Agricultural Sciences, No. 4 Xueyuan Road, Longhua, Haikou 571101, China; 2Key Laboratory of Vegetable Biology of Hainan Province, Institute of Vegetables Hainan Academy of Agricultural Sciences, No. 14 Xingdan Road, Qiongshan, Haikou 571100, China; 3Proving Ground, Chinese Academy of Tropical Agricultural Sciences, Baodao New Village, Nada Town, Danzhou 571737, China

**Keywords:** watermelon, tetraploid, morphology, photosynthetic parameters, proteomics, qRT-PCR

## Abstract

Watermelon is an important fruit that is widely distributed around the world. In particular, the production and consumption of watermelon in China ranks first in the world. Watermelon production is severely affected by a variety of biotic and abiotic stresses during cultivation, and polyploidization can promote stress resistance and yield. However, the morphological and physiological characteristics of tetraploid watermelon and the underlying molecular mechanisms are still poorly understood. In this study, we revealed that the leaves, fruits, and seeds of tetraploid watermelon were significantly larger than those of the diploid genotype. Some physiological characteristics, including photosynthetic rate (Pn) and stomatal conductance (Gs), were greater, whereas the intercellular CO_2_ concentration (Ci) and transpiration rate (Tr) were lower in tetraploid than in diploid watermelon. Two-dimensional gel electrophoresis combined with tandem mass spectrometry (MALDI-TOF/TOF) was performed to compare proteomic changes between tetraploid and diploid watermelon. A total of 21 differentially expressed proteins were identified; excluding the identical proteins, 8 proteins remained. Among them, four proteins were upregulated and four were downregulated in tetraploid versus diploid genotypes. qRT-PCR results showed inconsistencies in gene expression and protein accumulation, indicating a low correlation between gene expression and protein abundance. Generally, this study extends our understanding of the traits and molecular mechanisms of tetraploid watermelon and provides a theoretical basis for watermelon polyploid breeding.

## 1. Introduction

Watermelon (*Citrullus lanatus* Thunb. Matsum and Nakai), the fifth most consumed fruit in the world, is widely cultivated, mainly in tropical and temperate regions. The global annual production of watermelon was approximately 101.62 million tons in 2021, and countries including China, Turkey, India, Brazil, Algeria, and Iran are the major producers. China is the largest watermelon producer in the world, with annual production reaching 60.247 million tons in 2021, accounting for 59.3% of global output (FAOSTAT 2021). Watermelon has become the most popular summer fruit in China.

Because of its unique flavor and nutrients, including sugar, fiber, vitamins, antioxidants (lycopene and b-carotene), amino acids (citrulline and arginine), and minerals [1], watermelon is popular with consumers. However, watermelon plants under cultivation are subjected to various biotic and abiotic stresses, including low and high temperatures, drought, waterlogging, high salinity, heavy metals, insect pests, and pathogenic organisms, which seriously reduce the yield and quality of the fruit. To overcome these difficulties, watermelon breeders expect to develop multi-resistant and high-yielding varieties.

Ploidy breeding is an efficient approach to improving crop genetics. After chromosome doubling, also known as whole genome duplication, yield and resistance of crops are improved. The biomass yield of the tetraploid maize germplasm obtained by intersubspecific hybrids (maize (*Zea mays* L.) × teosinte (*Z. mays* spp. *mexicana* (Schrad.) Kuntze.)) increased 14% compared to the diploid genotype [2]. It was shown that tetraploid (4×) meadow fescue (*Festuca pratensis* (Huds.) Darbysh.) boosted seed yield by 100% and dry matter by 26% compared to diploid cultivars [3]. Furthermore, induced by colchicine, the yield of tetraploid rubber dandelion (*Taraxacum kok-saghyz*) was improved 47.7% more than the diploid genotype [4]. Due to the gene dosage effect, polyploidization can also promote plant resistance. For instance, it was reported that chromosome doubling enhanced drought tolerance in *Lycium ruthenicum* [5]. Moreover, tetraploid volkamer lemon (*Citrus volkameriana* Tan. and Pasq.) displayed superior salinity tolerance compared to diploid plants [6]. Compared with autotetraploid (4×) citrus seedlings, nutrient deficiency caused more damage to the diploid genotype [7]. Similar to these plants, leaf area, chlorophyll content, chlorophyll fluorescence (Fv/Fm), fruit weight, rind thickness, and seed length and width significantly were increased in seven autotetraploid watermelons compared with diploid watermelons [8]. After treatment with 300 mM NaCl, rootstock-grafted tetraploid watermelon displayed greater leaf stomatal conductance and higher net photosynthetic and transpiration rates than rootstock-grafted diploid watermelon [9].

In this study, we utilized oryzalin to treat diploid watermelon and successfully obtained tetraploid plants. We subsequently examined the morphological and physiological characteristics of the tetraploid plants. Further, two-dimensional gel electrophoresis of proteins was employed to investigate the differentially expressed proteins between diploid and tetraploid watermelon, and the protein expression was verified by qRT-PCR. Our findings reveal the impact of chromosome doubling on watermelon phenotype, photosynthesis, and molecular mechanisms, and provide some guidance for ploidy breeding of watermelon.

## 2. Materials and Methods

### 2.1. Plant Materials and Growth Conditions

In this study, we used double haploid and tetraploid watermelons from the inbred watermelon line “FR-32-1B”, which is bred by the Tropical Crops Genetic Resources Institute, Chinese Academy of Tropical Sciences, China. The seeds were soaked in distilled water for 2 h, then broken with nail clippers and placed at a constant temperature of 30 °C for germination. After germination, the seedlings were sown in 50-hole plastic pots filled with growing media containing peat soil (33.3%), perlite (33.3%), and vermiculite (33.4%). When the seedlings grew to the 2 true-leaf stage, they were planted in the greenhouse. The plant spacing and row spacing were 0.5 m and 0.7 m, respectively, with 15 plants per plot, 3 repetitions, and single vine pruning, using standard horticultural procedures for cultivation.

### 2.2. Tetraploid Watermelon Induction by Oryzalin

Tetraploid watermelon was induced by oryzalin according to a previous report [10], with slight modifications. Briefly, a 50 mg/L solution of oryzalin was prepared, drops of which were dripped on the growth points twice after the cotyledons of watermelon seedlings unfolded; one drop contained about 0.05 mL, and seedlings treated with distilled water were used as control. At least 30 seedlings were used per treatment.

### 2.3. Polyploid Watermelon Chromosome Counting

Chromosome counting is an efficient method for identifying polyploidy. Referring to the method established by Xu et al., with slight modification [11], briefly, 2n and 4n watermelon root tips were excised at a length of 1 to 1.5 cm, treated with saturated para-dichlorobenzene for 3 h, then fixed with a stationary fluid (absolute ethanol:glacial acetic acid = 3:1) for 8 h at 4 °C. Subsequently, 1 mol/L hydrochloric acid was added and dissociated in a water bath at 60 °C for 8 m, and finally rinsed with sterilized distilled water 3 times. About 0.2 cm of the root meristem tissue was excised and placed on a glass slide, stained with acetic magenta, observed under a microscope, and photographed.

### 2.4. Flow Cytometry

Detecting the nuclear DNA content of polyploid watermelons is an effective way to identify watermelon ploidy. Based on the method described by Sari et al. [12], with some modifications, briefly, we excised pieces of leaves about 1 cm^2^ from different polyploid watermelons and cut them into pieces, then immersed the pieces into lysis buffer. The crude cell nucleus extract was filtered with a 300-mesh filter, and subsequently stained with DAPI and analyzed by flow cytometry. Leaves treated with distilled water were used as the control.

### 2.5. Guard Cell Number and Chloroplast Number in Guard Cells

We collected 10-day-old functional leaves of diploid and tetraploid plants and tore out the epidermis, dropped distilled water on a glass slide, put the epidermis on the slide and allowed it to spread, then observed and counted guard cells and chloroplasts with an optical microscope.

### 2.6. Determination of Leaf Morphology

In the fruit-setting stage, 10-day-old functional leaves were collected and scanned with a leaf area meter, and the data were analyzed with Photoshop.

### 2.7. Photosynthesis and Chlorophyll Content

In the fruit-setting period, 10-day-old functional leaves were selected for photosynthesis measurement at 9:00–11:00 a.m. with the LI-6400ET photosynthesis analyzer. We cut the leaves into pieces with scissors, took 0.2 g samples and soaked them in 80% acetone for 24 h in the dark, then measured the chlorophyll content using a spectrophotometer at wavelengths of 663 and 645 nm.

### 2.8. Watermelon Leaf Protein Extraction

Watermelon leaf protein was extracted using a phenol extraction method according to previous reports with some modifications [13]. Briefly, we collected 0.5 g samples of 10-day-old watermelon leaves, ground them into powder with a pestle in liquid nitrogen, then added a 1.0 mL extraction buffer (containing 0.1M Tris (pH 8.0), 5% sucrose, 2% SDS, 50 mM dithiothreitol (DTT), and 2% protein inhibitor) and mixed it at 4 °C for 10 min. Extracted protein was reduced and alkylated by the method of Fan [14], and protein was quantified using Bradford assay with BSA as the standard [15].

### 2.9. Two-Dimensional Gel Electrophoresis of Proteins

The GE Healthcare 2-D electrophoresis system was utilized for isoelectric focusing according to a previous report [16]. For this process, 400 µg protein samples were loaded into gradient strip gels and rehydrated for 16 h at room temperature. Isoelectric focusing was performed with the following steps: 100 V for 1 h, 250 V for 1 h, 500 V for 1 h, 1000 V for 1 h, 8000 V for 1 h, gradient step from 8000 to 48,000 V, 500 V for 1 h. Thereafter, balance solution I (10 g·L^−1^ DTT, 50 mmol·L^−1^ Tris-HCI (pH 8.8), 6 mol·L^−1^ urea, 30% glycerol, 20 g·L^−1^ SDS) and balance solution II (25 g·L^−1^ iodoacetamide (IAA), 50 mmol·L^−1^ Tris-HCl (pH 8.8), 6 mol·L^−1^ urea, 30% glycerol, 20 g·L^−1^ SDS) were used to equilibrate the strip 2 times for 15 min each time. For the second dimension we used SDS-PAGE gel electrophoresis (separation gel concentration 12.5%, thickness 1 mm), first at 2 W per strip at low power for 1 h, and then at 15 W per strip for 6 h. Staining was performed by the Coomassie brilliant blue R-250 method according to a previous report [17].

### 2.10. Protein Analysis by MALDI-TOF-MS/MS

The strips were scanned with the ImageScanner gel imaging system, and differentially expressed protein spots (*p* < 0.05) were analyzed by ImageMaster 2D Platinum 6.0 software (GE Healthcare). The differentially expressed proteins were cut and identified by tandem mass spectrometry (MALDI-TOF/TOF).

### 2.11. Generation of Protein–Protein Interaction Networks

All differentially expressed proteins compared with autotetraploid and diploid watermelon were submitted to the STRING database (https://string-db.org/ (accessed on 26 August 2022)) to generate a wider protein interaction map.

### 2.12. RNA Isolation and qRT-PCR

The 10-day-old leaves of diploid and tetraploid watermelon at the fruit-setting stage were used as materials, the total RNA of leaves was extracted with an RNA Pure Plant Kit containing DNase I (Beijing Tian Gen Biotech Co., Ltd., Beijing, China), and RNA was reverse transcribed into cDNA by a PrimerScript™ RT Reagent Kit with gDNA Eraser (Takara Biomedical Technology Co., Ltd., Beijing, China). qRT-PCR of candidate differentially expressed genes was performed on an ABI PRISM7300 using TB Green^®^ Fast qPCR Mix (Takara Biomedical Technology Co., Ltd., Beijing, China). The 2-ΔΔCT method was used to analyze the data to determine relative gene expression levels. Three replicates were set for each sample, *ClACTIN* was set as the internal reference gene, and the PCR reactions of the internal reference gene and target gene were completed in the same batch. The experiment was repeated 3 times.

### 2.13. Statistical Analysis

Data represent at least three replicates. Statistical data were analyzed using one-way analysis of variance (ANOVA) and Student’s *t*-test. Significant differences between tetraploid and diploid genotypes were determined at *p* < 0.05 or *p* < 0.01.

## 3. Results

### 3.1. Confirmation of Watermelon Polyploidy

Flow cytometry technology has been widely used for the detection of genome size and chromosome ploidy in horticultural plants [18,19]. Thus, to determine the polyploidy level of diploid and autotetraploid watermelon, we employed flow cytometry to detect the DNA content in our materials. We found that the watermelon showed a fluorescence peak of nucleic DNA at channel 100 in diploid material (Figure 1A), whereas the major peak of tetraploid nucleic DNA was at channel 200 (Figure 1B). To further confirm the chromosome doubling of autotetraploid watermelon, we performed chromosome counting to determine the number of chromosomes in diploid and autotetraploid plants. The chromosome number of diploid watermelon was 22 (2n = 2x = 22), while the chromosome number of the autotetraploid genotype was 44 (4n = 4x = 44). Moreover, we also found that the tetraploid genotype had more chloroplasts in the guard cells, whereas the diploid genotype had more guard cells. These findings clearly suggest that we successfully constructed a tetraploid watermelon that was morphologically distinct from the diploid variety.

### 3.2. Morphological Comparison of Diploid and Tetraploid Watermelon

Previous studies have shown that plant polyploidy can change some traits of plants, such as leaf and fruit size and plant height [12,20]. After whole-genome duplication, we found that the leaves, fruits, and seeds of the tetraploid genotype were significantly altered (Figure 2 and Table 1). Compared with diploid, tetraploid leaves were wider (by 11.9%), leaf area was larger (by 21.9%), and leaf shape index became smaller (by 12.1%), while leaf length was not statistically different (Figure 2A and Table 1). Further, tetraploid fruits were wider (by 8%), longer (by 4.7%), and heavier (by 17.2%) compared to diploid, but the diploid genotype had a larger fruit shape index (by 2.86%). Besides leaves and fruits, tetraploid seeds became longer (by 6.84%), wider (by 9.78%), and thicker (by 21.9%) than diploid seeds, whereas the diploid genotype had a larger grain shape index (by 3.0%).

### 3.3. Physiological Analysis of Diploid and Tetraploid Watermelon

Due to the difference in leaf morphology between diploid and tetraploid watermelons, we wanted to investigate whether their physiological characteristics were also different. We used an LI-6400ET photosynthesis analyzer to detect the photosynthetic parameters of diploid and tetraploid watermelons. As shown in Figure 3A, tetraploid had a higher photosynthetic rate (Pn) than diploid (*p* < 0.01). Stomatal conductance (Gs) was greater in tetraploid than diploid (*p* < 0.05; Figure 3B), whereas the tetraploid displayed lower intercellular CO_2_ concentration (Ci) and transpiration rate (Tr) in leaves than diploid (*p* > 0.05; Figure 3C,D). Furthermore, lower chlorophyll a and chlorophyll b contents were found in tetraploid than diploid (Figure 3E,F), which was not statistically significantly different (*p* > 0.05).

### 3.4. Protein Profile of Diploid and Tetraploid Watermelon

Approximately 21 protein spots were generated from the two-dimensional gel electrophoresis experiments using the leaf protein of diploid and tetraploid watermelon. Among them, at least nine protein spots were identical in both as observed by eye and spot intensity ranking (Figure 4A,B). Using ImageMaster software (GE Healthcare), it was found that eight stained spots from tetraploid watermelon had fold change >2 and *p*-value < 0.05 compared with diploid watermelons. Of these, four spots were upregulated and four were downregulated (Figure 4C).

### 3.5. Functional Analysis of Screened Proteins

The 21 spots with differential expression were identified by MALDI-TOF-MS/MS. Referring to the watermelon genome, excluding the identical proteins, eight proteins were finally obtained; four proteins were upregulated and four were downregulated. To understand the function of these differentially expressed proteins, we performed a watermelon gene bank search (http://www.cucurbitgenomics.org/ (accessed on 26 August 2022)) to annotate them (Table 2). Among the upregulated proteins, *Cla009752* is a chlorophyll a/b binding protein gene, *Cla016932* is a jasmonate-induced protein gene, *Cla010223* is a low-temperature inducible protein gene, and *Cla011786* is a thioredoxin m protein gene. Among the downregulated proteins, *Cla008848* is a phosphoribulokinase/uridine kinase protein gene, *Cla017092* is a carbonic anhydrase protein gene, *Cla001764* is a chlorophyll a/b binding protein 8 gene, and *Cla007717* is a cytochrome b6-f complex iron–sulfur subunit protein gene.

### 3.6. Protein Interaction Networks

To illustrate the relationships among differentially expressional proteins, we generated a protein interaction map with them utilizing the STRING database (Figure 5). We found binding and regulatory relationships between seven proteins (homologous proteins of *Cla016932* were not found in *Arabidopsis thaliana*): CAB1 (chlorophyll a/b binding protein 1 (CAB1, *Cla009752*), phosphoribulokinase (PRK, *Cla002691*), photosystem I light-harvesting complex gene 3 (LHCA3, *Cla001764*), thioredoxin M4 (TRX-M4, *Cla011786*), carbonic anhydrase 1 (CA1, Cla017092), photosynthetic electron transfer C (PETC, *Cla007717*), and At2g03440 (*Cla010223*).

### 3.7. Validation of Differentially Expressed Proteins by qRT-PCR

Five differentially expressed proteins were selected to perform qRT-PCR to detect the expression of genes (Appendix A). As shown in Figure 6, of the genes corresponding to the selected proteins, three genes, *Cla009752*, *Cla010223*, and *Cla011786*, exhibited similar expression patterns to the three corresponding proteins, while the other genes, *Cla007717* and *Cla008848*, showed the opposite expressional pattern. This may be caused by post-transcriptional, translational, or post-translational regulation.

## 4. Discussion

Colchicine was first found to be an inhibitor of meiosis in 1930 [21], and subsequently was widely used for chromosome doubling in various plants, such as *Platanus acerifolia* [11], *Brachiaria brizantha* [22], and *Zea mays* L. [23]. Noh et al. found the best effect when 0.2% colchicine was used to treat inverted hypocotyl [24]. In addition, herbicides such as pronamide, oryzalin, amiprophose (APM), and trifluralin have also been used in plant polyploidization [25]. Compared to colchicine, herbicides have lower lethality and higher induction rates, so they have been more widely utilized in chromosome doubling of cucurbit crops [26]. To obtain tetraploid watermelon material, we treated diploid watermelon seeds with a suitable concentration of oryzalin (data not published). Diploid watermelon chromosome doubling was confirmed by chromosome counting and flow cytometry. Moreover, we observed that tetraploid watermelon guard cells had more chloroplasts but fewer guard cells in the same leaf area unit by microscopic observation.

Generally, whole-genome duplication can alter many phenotypic characteristics in plants. After genome replication, the tetraploid of *Echinacea purpurea* (L.) had substantially larger pollen grains, stomata, flowers, and seeds. Furthermore, the chlorophyll content and the width and thickness of leaves were boosted in tetraploid plants [27]. Similarly, compared to the diploid of *Nicotiana alata*, the tetraploid genotype had longer stem length and greater stem diameter, leaf breadth, and leaf length [28]. Likewise, the length and width of leaves and floral organs were increased in the tetraploid of *Paulownia tomentosa* [29]. In addition, it was reported that replication of the genome in *Plantago psyllium* improved the pollen grains, spikes and seeds, leaf thickness, and plant height [30]. In watermelon, in addition to a larger fruit size, the leaves, stem width, and staminate flowers of tetraploid watermelon (CLT1) were larger than those of diploid plants (CLD1) [31]. In another study, it was found that chromosome doubling enhanced seed size, cotyledon size, stem thickness, and trichome density and length [32]. Furthermore, there were differences in lycopene and hormone content between tetraploid and diploid watermelon during the fruit development stage; tetraploid watermelon contained more lycopene and IAA than the diploid genotype [33].

Consistent with these findings, we found that tetraploid watermelons displayed significantly larger leaves, fruits, and seeds, even if their leaf length was shorter compared to diploid watermelons. Plant leaves are closely related to plant photosynthesis. Along with the variation of leaf characteristics in tetraploid plants, their photosynthetic capacity is also changed. The tetraploid of *Acer buergerianum* Miq. showed higher chlorophyll content as well as maximal photochemical efficiency and potential photochemical efficiency of PSII than the diploid genotype [34]. In *Dendranthema nankingense* (Nakai) Tzvel, the tetraploid genotype had higher chlorophyll (a + b) content than diploid plants [35]. Similar to these tetraploid plants, autotetraploid apple showed increased chlorophyll content in the leaves as well as photosynthetic efficiency and fluorescence performance [36]. Consistent with these findings, our studies show that tetraploid watermelons had more chlorophyll a and chlorophyll b content and higher photosynthetic parameters than the diploid genotype. These results support the supposition that chromosome doubling may enhance photosynthesis in leaves.

Proteins are the executors of life activities and key factors in determining plant traits [37,38]. Chromosome doubling affects gene translation and protein processing, thereby altering plant phenotype. Proteomics research has shown that the translation of multiple genes is changed in the tetraploid genotype compared with the diploid genotype [39,40]. For example, compared to diploid rice, tetraploid rice had more CAT proteins, peroxidase P7, and catalase isozyme A [41]. Autotetraploid *P. australis* exhibited a higher expression of protein genes related to glutathione metabolism, cell division, and chlorophyll, cellulose, and lignin synthesis than the diploid genotype [42]. Some photosynthesis genes and pathways, such as the cytochrome b6-f complex, F-type ATPase, the light-harvesting chlorophyll protein complexes, and photosystem I and II, are more enriched in tetraploid *Liriodendron sino-americanum* than the diploid genotype [43]. Cheng et al. reported increased photosynthesis, cell defense and rescue, and biosynthesis peptides in tetraploid *Solanum tuberosum* L. compared to the diploid genotype [44]. Interestingly, in line with these findings, we found that photosynthesis and stress response proteins such as chlorophyll a/b binding protein 21 and thioredoxin m were elevated in tetraploid watermelon. However, a few proteins were correlated with photosynthesis; for example, chlorophyll a/b binding protein 8 and cytochrome b6-f complex iron–sulfur subunit were downregulated. These results indicate that plant polyploidy affects photosynthesis and stress response by altering the expression of related proteins.

Gene expression and protein translation are two key intrinsic factors affecting plant phenotype, and both of them co-regulate plant traits. However, transcript expression and protein abundance are not positively correlated. It was reported that the fold changes in gene expression and protein accumulation were inconsistent in *Candidatus liberibacter asiaticus* infected ‘Madam Vinous’ sweet orange, and the correlation coefficient of the overall proteome and transcriptome data was very low [45]. In addition, it was revealed that there were many discordant fold changes in mRNA–protein pairs of *A. thaliana* roots and leaves [46,47]. Furthermore, two-thirds of differentially expressed proteins in allotetraploid *Brassica napus* compared to diploid progenitors showed different fold changes with differentially expressed transcripts [48]. These findings support the supposition that a low level of correlation between protein abundance and transcript expression is universal in plants.

In our studies, we carried out 2-D electrophoresis to screen for differentially expressed proteins between diploid and autotetraploid watermelon. Of the five detected proteins, the accumulation pattern of three proteins was consistent with the transcript expression, and the others were the opposite. These data indicate that alterations at the transcript level are just one important factor in the change of protein abundance. Several factors may contribute to the inconsistencies between protein abundance and RNA transcript levels. First, the process of gene expression and regulation is very complicated. Transcript levels are regulated by post-transcriptional regulation and modification; for example, small RNAs (microRNAs, small interfering RNAs, and phasiRNAs) cleave the mRNAs of target genes and repress protein translation [49,50,51]. RNA modification, such as m6A and m5C, accommodates mRNA expression and stability [52,53], and many mRNAs can generate more than one transcript by alternative splicing [54]. Second, the differences in two-dimensional electrophoresis and qRT-PCR experimental methods limit the correlation between mRNAs and proteins.

## Figures and Tables

**Figure 1 bioengineering-09-00746-f001:**
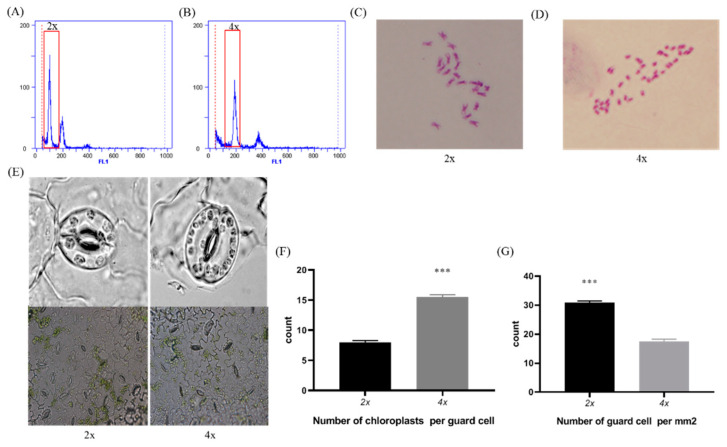
Flow cytometry analysis and chromosome counting of diploid and tetraploid watermelon. Relative nucleic DNA content of (**A**) diploid watermelon and (**B**) tetraploid watermelon; chromosome number of (**C**) diploid watermelon and (**D**) tetraploid watermelon; (**E**) stomata and chloroplast of guard cells in diploid and tetraploid plants; (**F**) number of chloroplasts per guard cell; (**G**) number of guard cells per mm^2^. Values are means ± SD (n = 3). Asterisks indicate statistically significant differences compared with diploid watermelon by Student’s *t*-test (*** *p* < 0.001).

**Figure 2 bioengineering-09-00746-f002:**
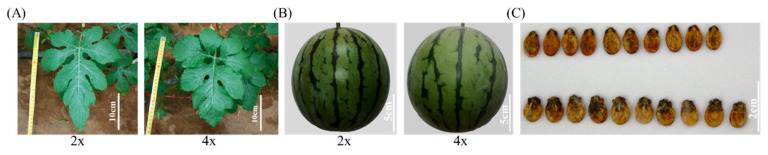
Morphological comparison between diploid and tetraploid watermelon: (**A**) plant leaf morphology; (**B**) fruit morphology; (**C**) seed morphology. Scale bar: 10, 5, and 2 cm, respectively.

**Figure 3 bioengineering-09-00746-f003:**
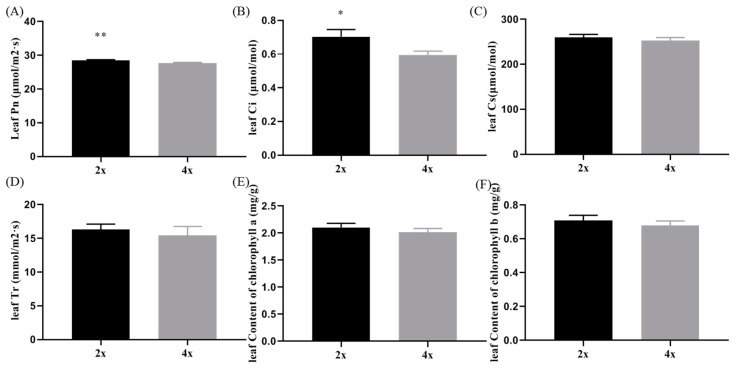
Comparison of photosynthetic parameters in diploid and tetraploid watermelon: (**A**) Pn; (**B**) Ci; (**C**) Cs; (**D**) Tr; (**E**) chlorophyll a content; (**F**) chlorophyll b content. Asterisks indicate statistically significant differences compared with diploid watermelon by Student’s *t*-test (* *p* < 0.05, ** *p* < 0.01).

**Figure 4 bioengineering-09-00746-f004:**
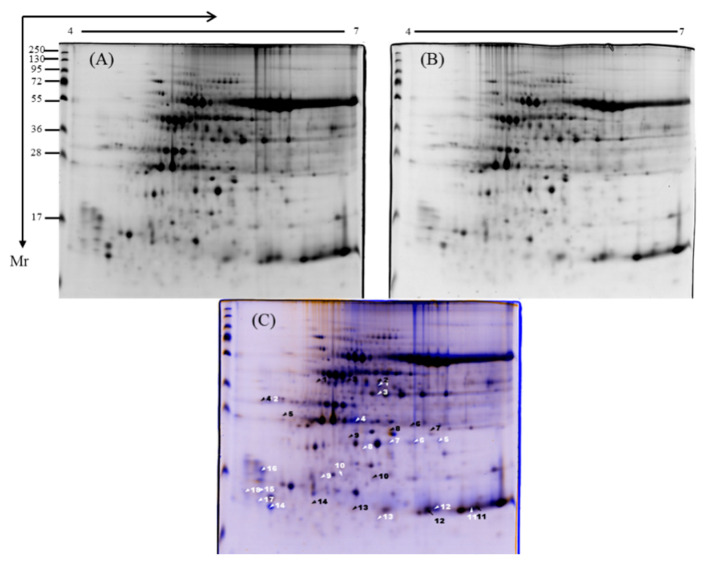
Two-dimensional electrophoresis map of diploid and tetraploid watermelon and screening of differential proteins. Proteome patterns of leaves in (**A**) diploid watermelon and (**B**) tetraploid watermelon; (**C**) 2-DE map showing differential proteins from two polyploid watermelons. White and black arrows indicate differentially expressed proteins of tetraploid compared with diploid watermelon (fold change >2, *p*-values < 0.05); white arrows indicate upregulated protein spots and black arrows indicate downregulated protein spots.

**Figure 5 bioengineering-09-00746-f005:**
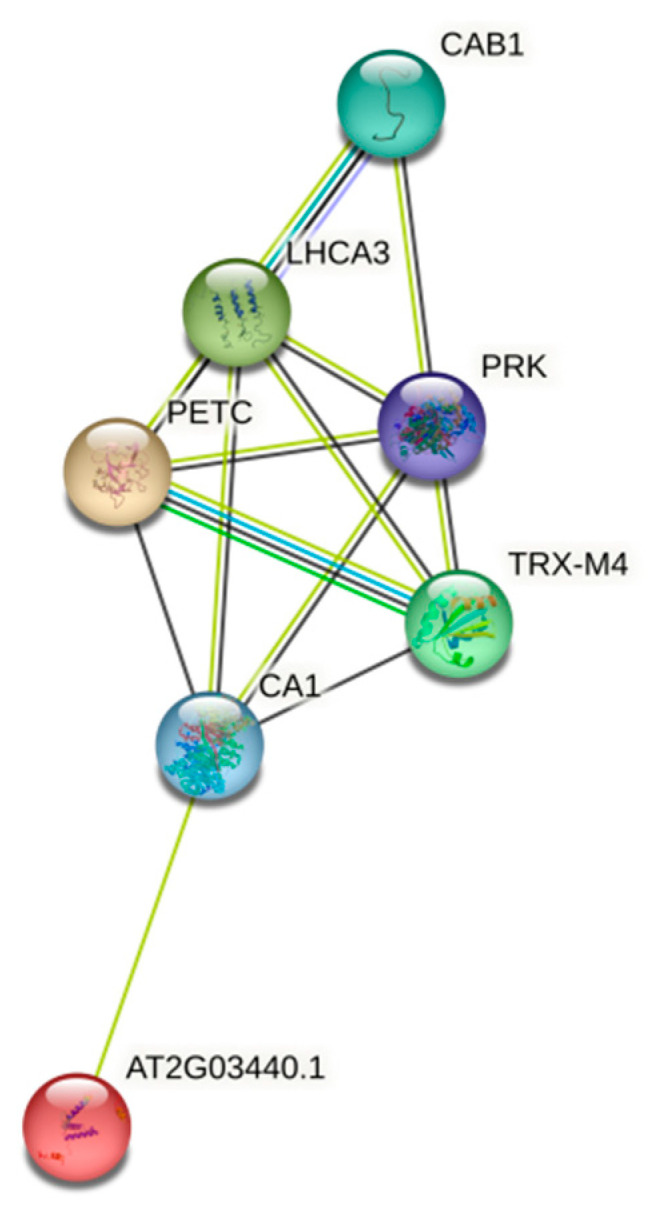
Protein interaction map of eight differential proteins (CAB1, PRK, LHCA3, TRX-M4, CA1, PETC, and At2g03440) established by employing STRING database.

**Figure 6 bioengineering-09-00746-f006:**
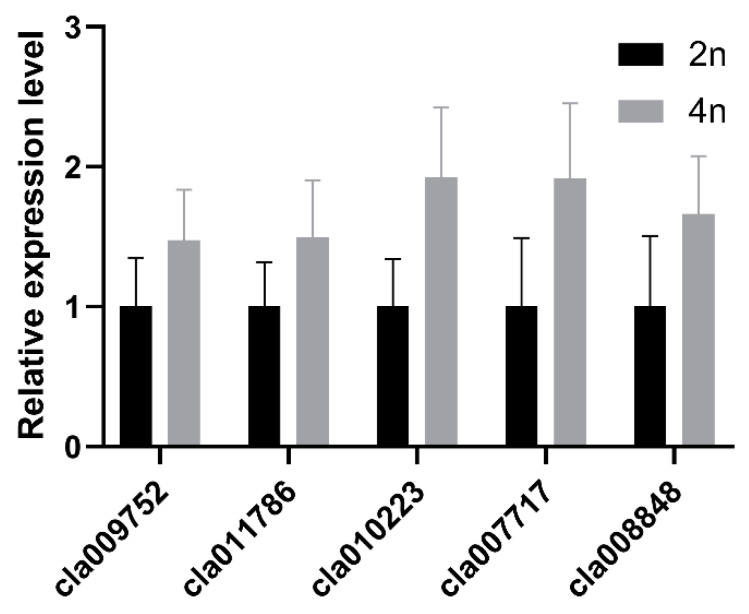
qRT-PCR analysis of five differentially expressed genes selected randomly between diploid and tetraploid watermelon. *Cla009752*, chlorophyll a/b binding protein 21; *cla011786*, thioredoxin m; *cla010223*, low-temperature inducible; *cla007717*, cytochrome b6-f complex iron–sulfur subunit; *cla008848*, phosphoribulokinase/uridine kinase. *ClACTIN* was used as endogenous control. Expression level of diploid was set to 1. Data are means ± SD of three replicates.

**Table 1 bioengineering-09-00746-t001:** Agronomic traits of diploid and tetraploid watermelon.

Ploidy Level	Leaf Length (cm)	Leaf Width (cm)	Leaf Area (cm^2^)	Fruit Length	Fruit Width	Fruit Weight	Seed Length	Seed Width	Seed Thickness
2×	26.29 ± 0.73	19.98 ± 0.90	223.50 ± 8.54	16.75 ± 1.31	15.95 ± 0.95	2.21 ± 0.39	9.06 ± 0.34	5.44 ± 0.13	2.10 ± 0.08
4×	26.01 ± 1.38	22.36 ± 0.25 *	272.50 ± 6.24 *	17.54 ± 1.56	17.23 ± 1.41 **	2.59 ± 0.59 *	9.68 ± 0.36 **	6.03 ± 0.41 **	2.56 ± 0.23 **

Asterisks indicate statistically significant differences compared with diploid watermelon by Student’s *t*-test (* *p* < 0.05, ** *p* < 0.01).

**Table 2 bioengineering-09-00746-t002:** Differentially expressed proteins between tetraploid and diploid watermelon.

Spot Number	Description	Gene ID ^a^	Biological Function	Theoretical pI /Mw (kDa)	Fold Change in Pairwise Comparison of 2x/4x	Sequence Coverage ^b^ /No. of Unique Peptides Matched ^c^
1	Chlorophyll a/b binding protein 21	*Cla009752*	photosynthesis	5.09/28.29	+3.167	17.4/4
2	Jasmonate-induced protein	*Cla016932*	unknown	6.39/23.90	+5.803	16.0/3
3	Low-temperature inducible	*Cla010223*	unknown	4.90/15.30	+∝	40.8/5
4	Thioredoxin m	*Cla011786*	stress/transport	9.32/19.50	+12.103	35.0/7
5	Phosphoribulokinase/uridine kinase	*Cla008848*	photosynthesis	8.19/38.85	-∝	5.8/2
6	Carbonic anhydrase	*Cla017092*	carbon utilization	6.71/36.33	-2.650	14.8/4
7	Chlorophyll a/b binding protein 8	*Cla001764*	photosynthesis	9.38/29.31	-2.654	8.4/2
8	Cytochrome b6-f complex iron–sulfur subunit	*Cla007717*	photosynthesis	7.77/28.22	-2.637	18.9/4

Values in table shown as mean ± SE of triplicate analysis; plus sign indicates upregulation and minus sign indicates downregulation. ^a^ Gene ID corresponding to the protein. ^b^ Percentage of identified peptide sequences in all protein sequences. ^c^ Number of unique peptides identified by MALDI-TOF-MS/MS matching the protein sequences.

## Data Availability

The data in this study are available within the figures and tables of this paper.

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
