# Peer review of "A Comparative Study of Morphology, Photosynthetic Physiology, and Proteome between Diploid and Tetraploid Watermelon (*Citrullus lanatus* L.)"

_bioengineering, 2022, doi:10.3390/bioengineering9120746_

Round 1
Reviewer 1 Report
Need further improvement
The study design and method are sound but the results and discussion need further improvement.
Some statements do not make sense too :
Watermelon is one of the most important fruits in the world and widely distributes around the world.
To understand the function of eight differentially expressed proteins, we performed watermelon gene banks (http://www.cucurbitgenomics.org/) 244 to annotate these proteins (Table 2).
This figure needs to be redrawn. Figure 6. qRT-PCR analysis of five differentially expressed genes selected randomly between dip-287 loid and tetraploid watermelon.
Author Response
Dear editors and reviewers,
Thank you very much for further reviewing our manuscript. We highly appreciate the editors and reviewers for their careful reading, constructive suggestions, and comments on our manuscript, which were very helpful for us to improve the manuscript. We have revised the manuscript according to the comments and suggestions. In the revision, we have carefully proofed the text for English writing. We have also addressed other issues raised by the other reviewers, please find our point-by-point answers and responses and the revised manuscript marked highlight. As suggested, we are now submitting the revised version of the manuscript.
Your input indeed greatly improves our manuscript, thank you very much in this regard.
Sincerely,
Huang He, Ph.D.
Tropical Crops Genetic Resources Institute
Chinese academy of Tropical Agricultural Sciences
Haikou 571101, China
Tel: +86-13648648159
E-mail: huanghe@catas.cn
Response to Reviewer 1 Comments
Point 1: The study design and method are sound but the results and discussion need further improvement.
Response 1: Thank you for the comments. In response to your suggestions, we have carefully proofed the text for English writing. We also rephrased the results and discussion and added the comparative analyses and conclusion in the results. We appreciate all the suggestions to polish the manuscripts.
Point 2: Some statements do not make sense too:
Watermelon is one of the most important fruits in the world and widely distributes around the world.
Response 2: Thanks for this suggestion. In the revision, we have rewritten this sentence in the Abstract section as follows: “Watermelon is an important fruit that is widely distributed around the world”. Please see Line 17 in the revised manuscript.
Point 3: To understand the function of eight differentially expressed proteins, we performed watermelon gene banks (http://www.cucurbitgenomics.org/) 244 to annotate these proteins (Table 2).
Response 3: Thanks for this suggestion. In the revision, we revised it as “To understand the function of these differentially expressed proteins, we performed a watermelon gene bank search (http://www.cucurbitgenomics.org/) to annotate them (Table 2)”. Please see Lines 248-250 in the revised manuscript.
.
Point 4: This figure needs to be redrawn. Figure 6. qRT-PCR analysis of five differentially expressed genes selected randomly between dip-287 loid and tetraploid watermelon.
Response 4: Thanks for the comment. We've solved the problem of garbled code in the original figure and have redrawn the new Figure 6 in the revised manuscript.

Reviewer 2 Report
The differences between polyploid and diploid plants need further scientific research, so the topic of this paper is very valuable. However, there are several problems in this paper that need to be considered by the author.
1, In the third paragraph of the introduction, many benefits of polyploidy are mentioned. Is there no previous literature on the differences in morphology, photosynthetic efficiency and physiological characteristics between polyploid and diploid? The author should give a detailed overview and summary in this regard.
2, Has anyone studied tetraploid watermelon before? The author should refer to and discuss.
3, Should it be "4. n" or "4n" in line 88?
4, Figures 5 and 6 are of poor quality, and there are still garbled codes?
Author Response
Dear editors and reviewers,
Thank you very much for further reviewing our manuscript. We highly appreciate the editors and reviewers for their careful reading, constructive suggestions, and comments on our manuscript, which were very helpful for us to improve the manuscript. We have revised the manuscript according to the comments and suggestions. In the revision, we have carefully proofed the text for English writing. We have also addressed other issues raised by the other reviewers, please find our point-by-point answers and responses and the revised manuscript marked highlight. As suggested, we are now submitting the revised version of the manuscript.
Your input indeed greatly improves our manuscript, thank you very much in this regard.
Sincerely,
Huang He, Ph.D.
Tropical Crops Genetic Resources Institute
Chinese academy of Tropical Agricultural Sciences
Haikou 571101, China
Tel: +86-13648648159
E-mail: huanghe@catas.cn
Response to Reviewer 2 Comments
Point 1: The differences between polyploid and diploid plants need further scientific research, so the topic of this paper is very valuable. However, there are several problems in this paper that need to be considered by the author.
1, In the third paragraph of the introduction, many benefits of polyploidy are mentioned. Is there no previous literature on the differences in morphology, photosynthetic efficiency and physiological characteristics between polyploid and diploid? The author should give a detailed overview and summary in this regard.
Response 1: Thank you very much for the comments. We have added detailed overview and summary in the Introduction section to include previous studies on morphology, photosynthetic efficiency, and physiological characteristics between polyploid and diploid watermelon. Please see Lines 65-70 in the revised manuscript.
Point 2: Has anyone studied tetraploid watermelon before? The author should refer to and discuss.
Response 2: Thank you very much for the suggestion. We have rewritten the discussion and added the previous reports on tetraploid watermelon research (Lines 288-289 and 306-313).
Point 3: Should it be "4. n" or "4n" in line 88?
Response 3: Thank you very much for the suggestion. This is our negligence in writing, we have revised “4.n” as “4n” in the revised version.
Point 4: Figures 5 and 6 are of poor quality, and there are still garbled codes?
Response 4: Thanks for the comment. We have redrawn the Figure 5 and 6 with high resolution , and resolved the problem of garbled codes.

Round 2
Reviewer 2 Report
Authors have addressed all comments.